# Redox-triggered switching in three-dimensional covalent organic frameworks

Chao Gao [1,4], Jian Li[2,3,4], Sheng Yin [1], Junliang Sun [2,3✉] & Cheng Wang [1✉]

The tuning of molecular switches in solid state toward stimuli-responsive materials has attracted more and more attention in recent years. Herein, we report a switchable three-dimensional covalent organic framework (3D COF), which can undergo a reversible transformation through a hydroquinone/quinone redox reaction while retaining the crystallinity and porosity. Our results clearly show that the switching process gradually happened through the COF framework, with an almost quantitative conversion yield. In addition, the redox-triggered transformation will form different functional groups on the pore surface and modify the shape of pore channel, which can result in tunable gas separation property. This study strongly demonstrates 3D COFs can provide robust platforms for efficient tuning of molecular switches in solid state. More importantly, switching of these moieties in 3D COFs can remarkably modify the internal pore environment, which will thus enable the resulting materials with interesting stimuli-responsive properties.

[1] Sauvage Center for Molecular Sciences and Key Laboratory of Biomedical Polymers (Ministry of Education), College of Chemistry and Molecular Sciences, Wuhan University, Wuhan 430072, China. [2] College of Chemistry and Molecular Engineering, Beijing National Laboratory for Molecular Sciences, Peking University, Beijing 100871, China. [3] Department of Materials and Environmental Chemistry, Stockholm University, Stockholm 10691, Sweden. [4] These authors contributed equally: Chao Gao, Jian Li. ✉email: junliang.sun@pku.edu.cn; chengwang@whu.edu.cn

Molecules that can reversibly change their structures/ properties in response to the external stimuli[1,2] (e.g., light, pH, heat, redox reagents, etc.) have received intensive attention, due to their important roles in the development of smart molecular devices for drug delivery[3,4], biosensor[5,6], information storage[7,8] and so on. Although most efforts had been focusing on studying their switching behaviors in solution state, the incorporation of these molecular switches into solid-state material toward promising applications has attracted more and more interests in recent years[9,10]. Unfortunately, their solution-state chemistry does not always translate into solid state[11], presumably owing to the spatial confinement effect. In this regard, porous materials have provided a promising platform to preserve their switchable functions in solid state, as the pore structure can offer the essential space for transformation. For example, metal-organic frameworks (MOFs) have shown the ability in maintaining the property of molecular switches in solid state, by immobilizing them into the framework as organic components[12–14]. However, it should be mentioned here, the moderate stability of MOFs will be a fatal obstacle to the development of this field[13], especially after considering their practical use as stimuli-responsive porous materials.

Covalent organic frameworks (COFs)[15,16], an emerging class of porous crystalline polymers with two- or three-dimensional (2D or 3D) structures, have gained increasing attention and found potentials in gas storage and separation[17–21], optoelectronics[22–26], heterogeneous catalysis[27–30], sensor[31–34], energy storage[35–38], etc. In principle, COFs can also be used as the platform for efficient tuning of molecular switches in solid state, but more importantly, they can show enhanced stability in contrast to most MOFs, due to their robust covalent linkages in the framework. There are several attempts to construct such systems and very few stimuli-responsive COFs have been announced[39–42]. For example, Jiang et al. reported an anthracene-based 2D COF, which can show structural transformations by alternating UV light irradiation and heating[38]. Trabolsi et al. reported an azobenzene-equipped 2D COF that can be used as light-operated reservoir[41]. Unfortunately, although these systems have shown interesting stimuli-responsive properties, there still has some problems that need to be further improved, such as inefficient conversion. These drawbacks may probably be explained by their inherent 2D structures, as the strong π–π interaction in the adjacent layers can strongly restrict the transformation of molecular switches.

3D COFs, in which the molecular building blocks are three-dimensionally linked to form the extended network, can characteristically possess numerous isolated sites and abundant open channels[43,44]. Theoretically, these unique advantages can minimize the interference between individual switches and also provide enough space for their conformational change, which will enable 3D COFs more suitable than 2D COFs for switching molecules in solid state. However, due to the synthetic difficulty and complicated structural determination, the construction of 3D COFs has been considered as a big challenge[45]. Since first reported by Yaghi in 2007[43], only a handful (~50 examples) of 3D COFs have been announced[46–57]. In addition, their applications are still limited and need to be further explored[44]. Therefore, the construction of molecular switches based 3D COFs toward stimuli-responsive applications is highly demanded.

Herein, we report the design and synthesis of a switchable 3D COF (3D-TPB-COF-HQ), which can undergo a reversible transformation in the skeleton through a hydroquinone/quinone redox reaction (Fig. 1). Our results clearly show that 3D-TPB-COF-HQ has high crystallinity and large surface area, and by using continuous rotation electron diffraction (cRED) technique[58,59], 3D-TPB-COF-HQ is determined to adopt a five-fold interpenetrated **pts** topology with a high-resolution cRED database of ~1.0 Å. Interestingly, 3D-TPB-COF-HQ can be oxidized into 3D-TPB-COF-Q and then reduced back to the initial state through redox reaction, with retaining the crystallinity and porosity. More importantly, due to the modification of pore environment during the transformation process, 3D-TPB-COF-Q exhibits a much higher $CO_2/N_2$ selectivity compared to 3D-TPB-COF-HQ, indicating a remarkable stimuli-responsive separation effect.

## Results

**Synthesis and characterization of 3D-TPB-COF-HQ.** In order to build a stimuli-responsive 3D COF containing molecular switch, we decided to choose the redox hydroquinone/quinone system as a typical example. Based on our reported topology design to construct 3D COFs[46,47], we designed and synthesized a linker TPB-HQ (Supplementary Fig. 1), which can react with tetra(p-aminophenyl)methane (TAPM) through [4 + 4] imine condensation reactions to form 3D-TPB-COF-HQ (Fig. 1). The formation of imine bonds was assessed by Fourier transform infrared (FT-IR) spectroscopy and solid-state NMR (ssNMR) spectroscopy. FT-IR spectrum showed a stretching vibration band at ~1624 $cm^{-1}$ (Supplementary Fig. 4) and the ssNMR spectrum exhibited a signal at ~160 ppm (Supplementary Fig. 5), strongly confirming the formation of imine bonds in 3D-TPB-COF-HQ. From thermogravimetric analysis (Supplementary Fig. 6), 3D-TPB-COF-HQ is thermally stable up to 500 °C. According to scanning electron microscopy images, 3D-TPB-COF-HQ has a granular morphology (Supplementary Fig. 7a, b). Moreover, the permanent porosity of 3D-TPB-COF-HQ was measured by nitrogen sorption isotherm at 77 K (Supplementary Fig. 8a). The Brunauer–Emmett–Teller (BET) surface area was calculated to be 842 $m^2 g^{-1}$ and by using nonlocal density functional theory (NLDFT), the calculated pore size distribution displayed a major peak at 0.52 nm (Supplementary Fig. 8b). Moreover, we studied the stability of 3D-TPB-COF-HQ in different solvents (DMF, DMSO, $CH_3CN$, MeOH, $H_2O$, 0.1 M HCl) for 24 h. As shown in Supplementary Fig. 9a, 3D-TPB-COF-HQ is stable in these solvents.

The powder X-ray diffraction (PXRD) pattern of 3D-TPB-COF-HQ showed amounts of intense diffraction peaks, indicating long-range ordering (Fig. 2a). As the cRED technique has recently shown the ability in determining the crystal structure of 3D COFs[47], we collected six individual cRED datasets on different 3D-TPB-COF-HQ crystals with a size of ~500 nm on JEM2100 TEM at 99 K by continuously rotating the goniometer. A monoclinic unit cell ($a = 28.75$ Å, $b = 7.88$ Å, $c = 25.33$ Å, $\alpha = 90°$, $\beta = 90.8°$, $\gamma = 90°$) and possible space groups ($C2/c$ and $Cc$) were obtained from a typical single crystal diffraction dataset by reconstructing the 3D reciprocal lattice using REDp data procession software (Fig. 2b)[59]. It should be mentioned here, the obtained typical cRED dataset has a high resolution of 1.0 Å (Supplementary Fig. 10), which allows us to solve the structure of 3D-TPB-COF-HQ on atomic resolution level. The structure solution was conducted on the merged datasets through the software *SHELXT*, and all non-hydrogen atomic positions in the framework are located by different electrostatic potential map with the space group of $C2/c$. The 3D-TPB-COF-HQ is identified to adopt a five-fold interpenetrated **pts** topology (Fig. 3). The Rietveld refinement with rigid-body constraints was further performed on the experimental PXRD of 3D-TPB-COF-HQ. The $R_{wp}$ and $R_p$ of the final Rietveld refinement were converged to 4.74% and 3.40% with the unit cell of $a = 28.761 (9)$ Å, $b = 7.645 (1)$ Å, $c = 25.535 (3)$ Å, $\alpha = \beta = 90°$, and $\gamma = 94.720 (9)°$ (Supplementary Tables 1–2).

**Redox-triggered transformation in 3D-TPB-COF-HQ.** Due to the existence of hydroquinone units, the redox-triggered

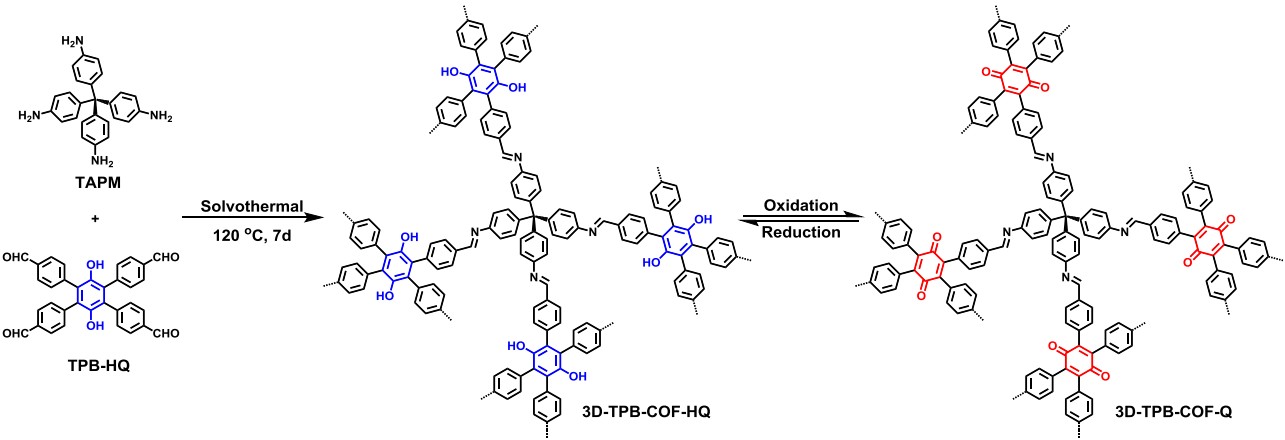

**Fig. 1 Chemical Structure.** Schematic representation of the synthesis and redox-triggered switching behavior of 3D-TPB-COF-HQ.

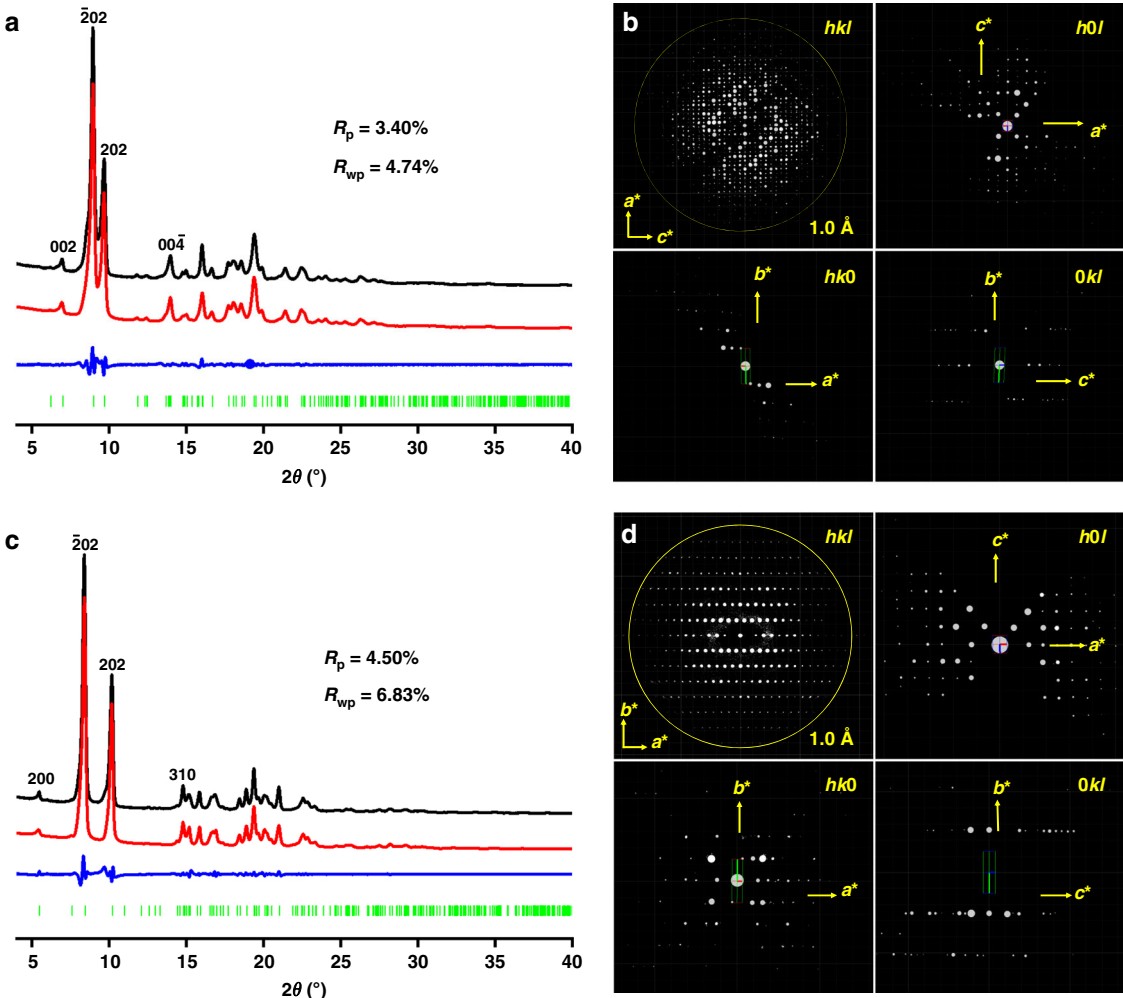

**Fig. 2 The Rietveld refinement and 3D reciprocal lattice.** The Rietveld refinement of 3D-TPB-COF-HQ (**a**) and 3D-TPB-COF-Q (**c**): the experimental XRD patterns are shown in black, the Rietveld refinement patterns in red, and their difference in blue, Bragg position from the five-fold interpenetrated **pts** net in green. 3D reciprocal lattice of 3D-TPB-COF-HQ (**b**) and 3D-TPB-COF-Q (**d**). The same reflection conditions ($hk0$: $h + k = 2n$; $h0l$: $h = 2n$, $l = 2n$) of 3D-TPB-COF-HQ and 3D-TPB-COF-Q was obtained by $h0l$, $0kl$, and $hk0$ slices cut from the 3D reciprocal lattice.

switching behavior of 3D-TPB-COF-HQ was investigated. For the oxidation process, we chose the benzoquinone as the oxidant (Fig. 4a). After addition of benzoquinone in $CH_3CN$ to 3D-TPB-COF-HQ for 90 min, the pale yellow powders changed to orange solids (named as 3D-TPB-COF-Q). According to the integration of characteristic protons in the $^1H$ NMR spectroscopy of the digested 3D-TPB-COF-Q (Supplementary Fig. 19), about 90% of hydroquinone units in 3D-TPB-COF-HQ were oxidized to the quinone state. We should mention here, the extension of oxidation time has no obvious improvement in the conversion yield,

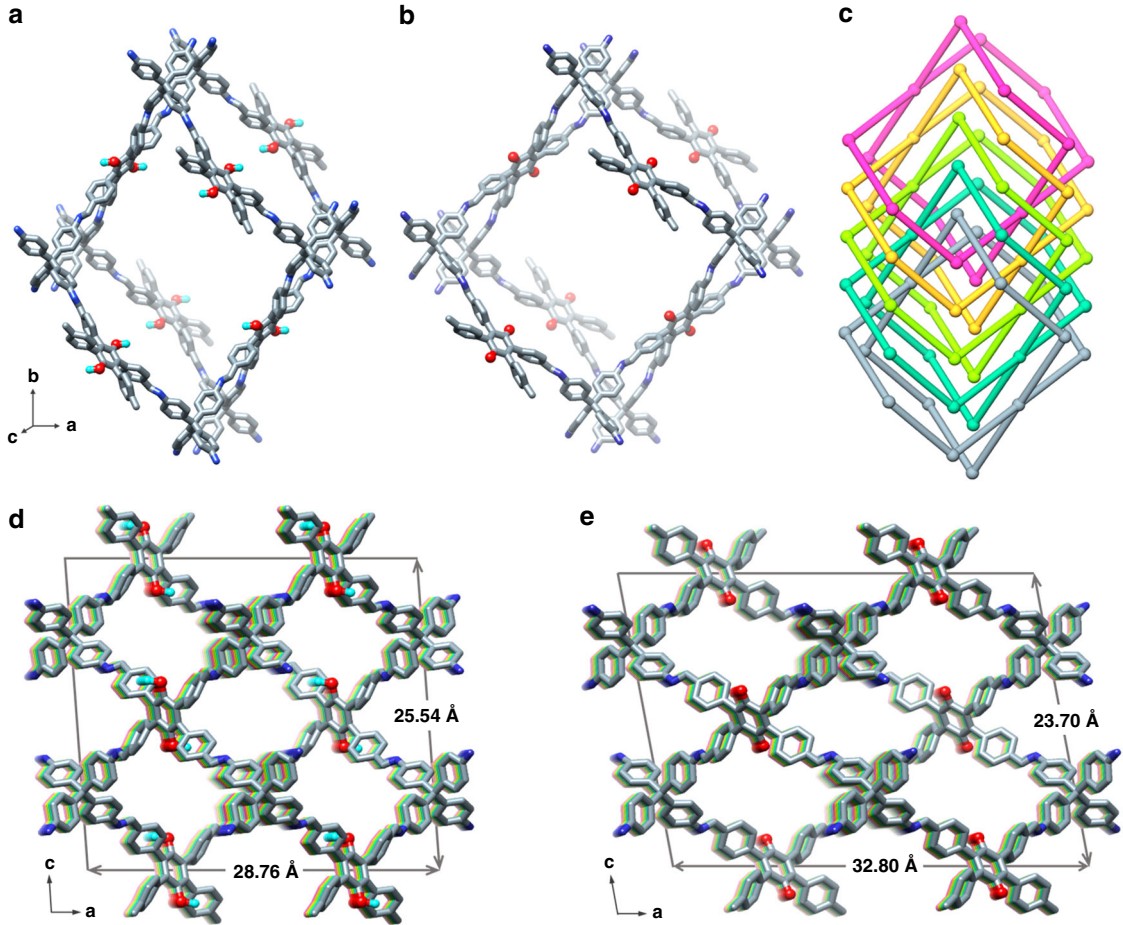

**Fig. 3 Structural representations. a** Single **pts** network of 3D-TPB-COF-HQ; **b** Single **pts** network of 3D-TPB-COF-Q; **c** five-fold interpenetrated **pts** topology; the porous structure of (**d**) 3D-TPB-COF-HQ and (**e**) 3D-TPB-COF-Q. The carbon is colored in gray and nitrogen is in blue; all the hydrogen atoms are omitted for clarity except the hydrogens in the central benzene rings of 3D-TPB-COF-HQ. The balls in 3,6-position of the central ring of TPB are the atoms: H, cyan; O, red.

which may be ascribed to the existence of some defects in 3D-TPB-COF-HQ. The PXRD pattern of 3D-TPB-COF-Q (Fig. 2c) also exhibited intensive peaks, but the peak positions are a little shifted compared to that of 3D-TPB-COF-HQ. For example, the strong peaks at 8.89° and 9.68° in 3D-TPB-COF-HQ moved to 8.45° and 10.22°, without much change of the relative intensity. This indicated that the unit cell parameters might have been changed but with the topology maintained. Although 10% hydroquinone units still existed, it has no obstacle for structure characterization on 3D-TPB-COF-Q by using cRED technique, because the powder sample can be treated as single crystals. Again, nano-sized individual 3D-TPB-COF-Q single crystals were selected to collect eight cRED datasets. Similar to the pristine sample, two of the datasets has high resolution of ~1.0 Å (Fig. 2d, Supplementary Fig. 12). After ab initio structural solution with the merged cRED dataset, 3D-TPB-COF-Q was confirmed to adopt a fivefold interpenetrated **pts** topology as expected. Furthermore, the structure of 3D-TPB-COF-Q can be refined against PXRD by the Rietveld refinement. The $R_{wp}$ and $R_{p}$ of final Rietveld refinement were converged to 6.83% and 4.50% with the unit cell of $a = 32.797$ (3) Å, $b = 7.199$ (1) Å, $c = 23.699$ (3) Å, $\alpha = \beta = 90°$, and $\gamma = 101.37$ (1)° (Supplementary Tables 3–4). Compared with 3D-TPB-COF-HQ, the channel shape of 3D-TPB-COF-Q changed with a little elongation along the $a$-axis while a little shrinkage along the $c$-axis (Fig. 3d, e). From the nitrogen sorption isotherm (Supplementary Fig. 8c), the BET

surface area of 3D-TPB-COF-Q was calculated to be 840 m² g⁻¹ and the calculated pore size distribution centered at 0.52 nm (Supplementary Fig. 8d), which were similar with that of 3D-TPB-COF-HQ. In addition, 3D-TPB-COF-Q also showed high stability in different solvents for 24 h (Supplementary Fig. 9b).

In order to prove that the oxidation process happened within the COF skeleton, we recorded the ssNMR spectra and FT-IR spectra of 3D-TPB-COF-HQ after different oxidation times. As shown in Fig. 4b, the ssNMR spectra exhibited the appearance and continuous growth of a new peak centered at 186 ppm, which should be assigned to the characteristic of carbon atoms of C=O groups in quinone units. Similarly, a new peak appeared at 1650 cm⁻¹ in FT-IR spectra (Supplementary Fig. 18), which should be assigned to the stretch vibration of C=O groups in quinone units. In addition, we took the PXRD patterns of 3D-TPB-COF-HQ after different oxidation periods (Fig. 4c). Obviously, the PXRD peaks of 3D-TPB-COF-Q increased, while the peaks for 3D-TPB-COF-HQ deceased gradually and finally disappeared. From these experimental data, it can be concluded that the oxidation reaction gradually happened within the framework.

The reduction of 3D-TPB-COF-Q back to 3D-TPB-COF-HQ [named 3D-TPB-COF-HQ(R)] was then investigated, by using ascorbic acid as the reductant (Fig. 4a). After the addition of ascorbic acid in methanol to 3D-TPB-COF-Q for 90 min, the color of powders changed from orange back to pale yellow again.

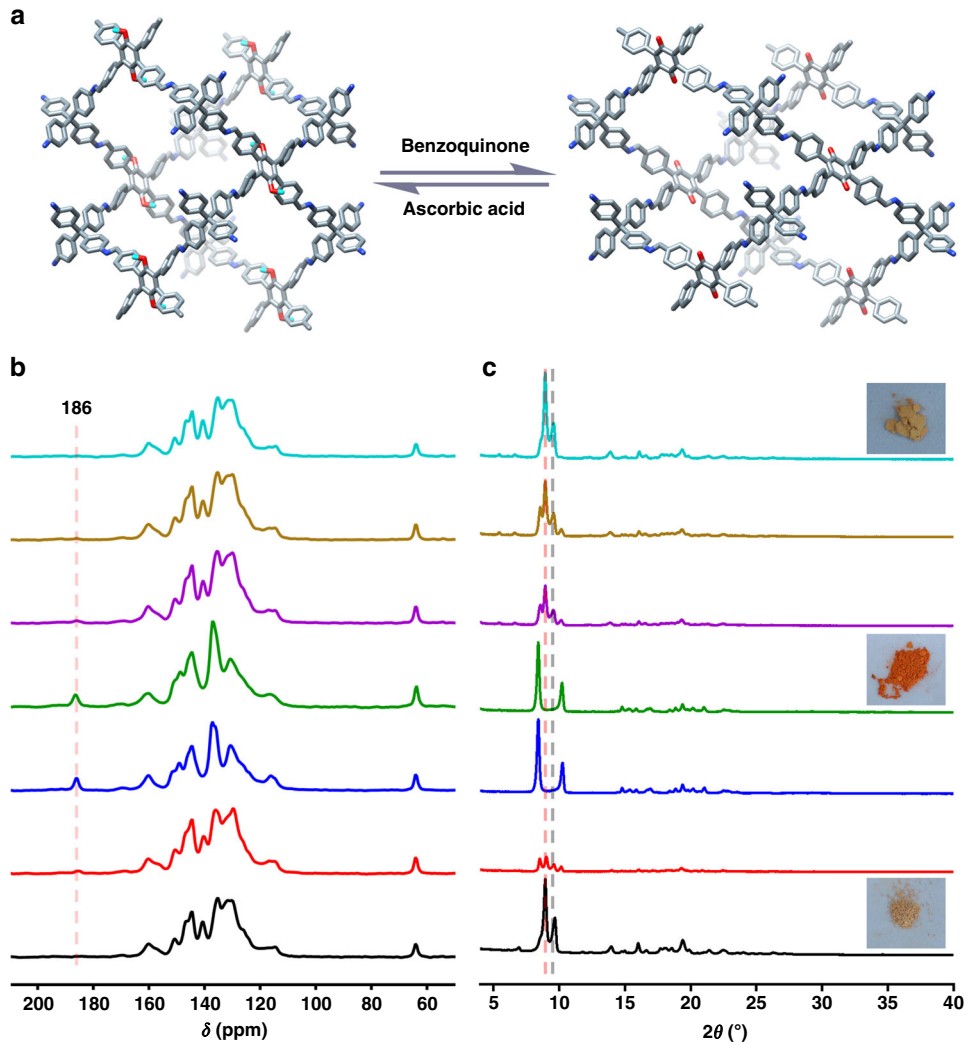

**Fig. 4 Redox-triggered switching process. a** Reversible crystal structures transformation between 3D-TPB-COF-HQ and 3D-TPB-COF-Q. **b** Solid-state $^{13}$C NMR spectra of 3D-TPB-COF-HQ during the oxidation/reduction process. (**c**) PXRD patterns of 3D-TPB-COF-HQ during oxidation/reduction process. Note: 3D-TPB-COF-HQ (black line); oxidation for 2 min (red line); oxidation for 40 min (blue line); oxidation for 90 min (3D-TPB-COF-Q, green line); reduction for 30 min (dark violet line); reduction for 60 min (sienna line); reduction for 90 min (3D-TPB-COF-HQ (R), cyan line). Inset: the pictures of corresponding materials.

By performing the $^1$H NMR spectroscopy of the digested 3D-TPB-COF-HQ(R), it can be concluded that all the quinone units in 3D-TPB-COF-Q were quantitatively reduced back to hydroquinone. In addition, the PXRD pattern of 3D-TPB-COF-HQ(R) matched well with that of 3D-TPB-COF-HQ. Furthermore, the BET surface area was calculated to be 816 m$^2$ g$^{-1}$ and the pore size distribution centered at 0.52 nm (Supplementary Fig. 8e, f), which is fitting well with 3D-TPB-COF-HQ. We also monitored this reduction process by performing ssNMR, FT-IR and PXRD experiment with different reduction times. As expected, the signal of quinone units in the ssNMR spectra (Fig. 4b) and FT-IR spectra (Supplementary Fig. 18) gradually disappeared, and the PXRD patterns (Fig. 4c) of 3D-TPB-COF-Q gradually changed back to the characteristic pattern of 3D-TPB-COF-HQ. Based on these results, 3D-TPB-COF-Q can be gradually reduced back to the original state through the framework. Therefore, 3D-TPB-COF-HQ and 3D-TPB-COF-Q could be reversibly converted to each other through oxidation/reduction reaction with the framework maintained. It should be mentioned here, this switching process can be repeated for at least three times (Supplementary Figs. 20–22), without losing crystallinity and porosity.

**Tunable gas absorption and separation.** Considering the redox-switchable process in 3D-TPB-COF-HQ will form different functional groups on the pore surface and also change the shape of the pore channel, we speculate the redox-triggered modification of pore environment may influence their gas separation property. For a proof-of-concept experiment, we studied the selective sorption of $CO_2$ over $N_2$ to demonstrate the redox-responsive effect. As shown in Fig. 5a, 3D-TPB-COF-HQ exhibited a $CO_2$ capacity of 93.4 cm$^3$ g$^{-1}$ at 273 K and 1 bar, and after oxidation to 3D-TPB-COF-Q, the capacity will increase a little bit to 105 cm$^3$ g$^{-1}$. We then calculated the isosteric heats of adsorption ($Q_{st}$) from the $CO_2$ adsorption isotherms at two different temperatures by fitting the data to the virial equation (Supplementary Table S5 and Supplementary Fig. 23-25). Obviously, the $Q_{st}$ values of 3D-TPB-COF-HQ (23.5 kJ mol$^{-1}$) is lower than that of 3D-TPB-COF-Q (29.0 kJ mol$^{-1}$). Based on the $CO_2$ and $N_2$ isotherms measured at 273 K and by using the ideal adsorbed solution theory (IAST), the adsorption selectivity for $CO_2/N_2$ mixtures (15:85) of these COFs as a function of pressure is also calculated (Fig. 5b). 3D-TPB-COF-HQ showed a selectivity of 40 at 1 bar, but after oxidation to 3D-TPB-COF-Q, the

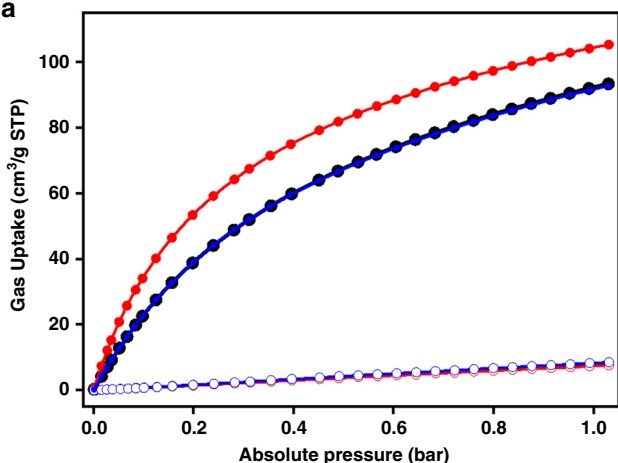

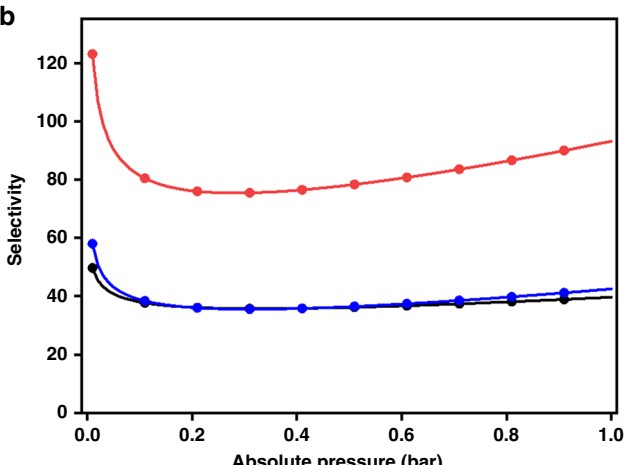

**Fig. 5 Redox-triggered gas separation. a** $CO_2$ and $N_2$ isotherms at 273 K: full symbols are for $CO_2$, empty symbols are for $N_2$; **b** IAST selectivity of $CO_2/N_2$ at the ratio of 15/85. Note: 3D-TPB-COF-HQ (black), 3D-TPB-COF-Q (red), 3D-TPB-COF-HQ(R) (blue).

selectivity will increase to a much higher value of 93. From these results, we believe that the shape change of the pore channel and the modification of pore surface from hydroquinone to quinone groups will significantly enhance the gas separation selectivity. Moreover, after the reduction of 3D-TPB-COF-Q, 3D-TPB-COF-HQ(R) exhibited almost the same performance as that of TPB-COF-HQ, confirming again the reversibility of this process. Therefore, the redox-triggered switching process in 3D-TPB-COF-HQ can reversibly change the pore environment and result in different gas separation property.

## Discussion

In summary, we have reported a stimuli-responsive 3D COF, which can undergo a reversible transformation through a hydroquinone/quinone redox reaction while retaining the crystallinity and porosity. Our results clearly showed that the redox reaction gradually happened through the framework, with an almost quantitative conversion yield. In addition, the atomic-level characterization of crystal structures by cRED techniques has provided detailed structural transformation in this process. Interestingly, this redox-triggered transformation in the framework can modify the pore environment, which will further lead to drastic changes in gas separation property. From this study, we strongly believe 3D COFs can provide an ideal platform for efficient tuning of molecular switches in solid state, by

immobilizing them into the framework. More importantly, their switching process can reversibly change the pore environments and thus endow the designed 3D COFs with interesting stimuli-responsive properties. Considering the increasing demands of stimuli-responsive porous materials, the construction of robust 3D COFs that can respond to different stimuli (e.g. light, temperature, etc.) for interesting applications can be imagined.

## Methods

**Synthesis of 3D-TPB-COF-HQ**. A Pyrex tube was charged with TAPM (30.4 mg; 0.08 mmol), 1,2,4,5-tetrakis-(4-formylphenyl)-3,6-dihydroxy-benzene (TPB-HQ) (42.0 mg, 0.08 mmol), mesitylene (7.2 mL), n-BuOH (0.8 mL) and 12 M aqueous acetic acid (0.8 mL). After being degassed by freeze-pump-thaw technique for five times and then sealed under vacuum, the tube was placed in an oven at 120 °C for 7 d. The resulting precipitate was filtered, exhaustively washed by Soxhlet extractions with tetrahydrofuran, methanol, and dichloromethane for 4 d, dried at 120 °C under vacuum for 12 h. The 3D-TPB-COF-HQ was isolated as pale yellow powder (52.9 mg, 79% yield). Elemental analysis for the calculated: C, 84.87%; H, 4.59%; N, 6.71%. Found: C, 80.11%; H, 4.58%; N, 6.62%.

**Synthesis of 3D-TPB-COF-Q**. A Pyrex tube was charged with 3D-TPB-COF-HQ (20.0 mg), p-benzoquinone (21.8 mg) and CH$_3$CN (5.0 mL). The tube was sealed under air and stirred for 90 min at 60 °C. The resulting precipitate was filtered and exhaustively washed by CH$_3$CN, dried at 120 °C under vacuum for 12 h. The 3D-TPB-COF-Q was isolated as orange powder (18 mg, 90%). From the $^1$H NMR spectrum of digested 3D-TPB-COF-Q, the content of quinone is about 90% (see Supplementary Method 7 for details). It should be emphasized here, we have tried to directly synthesize 3D-TPB-COF-Q from TAPM and TPB-Q many times, but unfortunately failed.

**Further reduction of 3D-TPB-COF-Q**. 3D-TPB-COF-Q (20 mg) was added to a tube containing a solution of ascorbic acid in MeOH (5.0 mL, 0.2 mol L$^{-1}$). After that, the mixture was stirred at 30 °C for 90 min. The resulting 3D-TPB-COF-HQ (R) (18.4 mg, 92%) was filtered and exhaustively washed by MeOH and DCM, dried under vacuum. From the $^1$H NMR spectrum of digested 3D-TPB-COF-HQ (R), all of quinone units were reduced to hydroquinone groups (see Supplementary Method 7 for details).

## Data availability

All the data supporting the findings of this study are available within the Article and its Supplementary Information, or from the corresponding author (C.W. or J.S.) upon reasonable request.

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

## Acknowledgements

C.W. gratefully acknowledges financial support from the National Natural Science Foundation of China (21975188 and 21772149), the Funds for Creative Research Groups of Hubei Province (2017CFA002) and the Fundamental Research Funds for Central Universities. J.S. acknowledges financial support from the National Natural Science Foundation of China (21871009, 21621061, and 21527803), the Swedish Research Council (VR), and the Knut and Alice Wallenberg Foundation (KAW) for financial supports. We also acknowledge Prof. Anmin Zheng and Dr. Xianfeng Yi for their help in recording ssNMR data.

## Author contributions

C.G. performed the synthesis and the characterizations of 3D COFs, including NMR, PXRD, FT-IR, gas absorption. S.Y. assisted with the experiment. J.L. and J.S. perfomred cRED experiment and solve the crystal structure. C.W. and J.S. supervised the experiment. C.W. designed the project and wrote the manuscript with the assistance of other authors.

## Competing interests

The authors declare no competing interests.
