## [Peer Review File · Nature Communications]

REVIEWER COMMENTS

Reviewer #1 (Remarks to the Author):

Overall, this is a very interesting and novel manuscript that would be of interest to the readers of Nature Communications. Therefore, I recommend it for publication after the authors address the following:

1. Figure S1 depicts the FT-IR of both 3D-TPB-COF-HQ and 3D-TPB-COF-Q. Since this figure is used to demonstrate the formation of the imine-linked framework, I believe it would be beneficial to include the FT-IR spectra of the two monomers to further drive home the successful imine condensation. This figure is also used to observe the oxidation of 3D-TPB-COF-HQ to 3D-TPB-COF-Q, as noted by the appearance of a carbonyl stretch in the oxidized product. The spectra are cut off and have an upward bound of ~2500 wavenumbers. I believe it would be best if the O-H region was included to show the initial presence of O-H groups in the reduced form and their disappearance upon oxidation (this should also be applied to Figure S12).
2. Did the authors try any other oxidizing/reducing agents? Why did they use a different oxidizer for the polymer [BQ] vs. the small molecule [Fe(ClO₄)₃]?
3. Since the authors mention that one advantage to using COFs over MOFs is the increased chemical stability of COFs, I believe that stability studies of the two synthesized 3D-COFs should be included in the SI.
4. In the manuscript, the authors claim that they are able to repeat the switching process three times “without losing crystallinity and porosity.” While they include data to support the retention of crystallinity, they do not include any data regarding the porosity of the material after repeated redox-switching.

Reviewer #2 (Remarks to the Author):

The manuscript submitted to Nat. Commun. by Gao et al. reports the synthesis a 3-dimensional covalent organic framework (COF) and its redox switching property by addition of oxidizing/reducing agent. The authors have tried to show the existence of the two forms through XRD analysis and advanced technique like electron diffraction.

1. In the introduction section, they highlighted that COFs are superior materials than MOFs in terms of chemical stability, which is partly true. There are certain MOFs which are quite stable and the same group has already reported this redox switching phenomenon using a chemically stable Zr-based MOF (Chem. Mater. 2015, 27, 18, 6426-6431). The chemical stability of imine-based COFs is also not good, owing to the reversible nature of the condensation reaction.
2. Added to that, switching between hydroquinone/quinone moieties inside a 2-dimensional COF is already reported (Chem. Mater. 2017, 29, 5, 2074-2080), using more facile approach of electrochemistry. Although this communication attempted to emphasize on introduction of 3-dimensional framework for better applicability, 5-fold interpenetration of the nets leaves with minimal porosity at the end.
3. Do the materials show any distinct emission property? If so, then monitoring the emission properties could be used to study the conversion kinetics between quinone/hydroquinone forms of

the framework.

4. There is slight difference between adsorption and desorption curve for N₂ adsorption isotherm at 77 K for the hydroquinone moiety containing framework, but not observed with the quinone form.

What could be the origin for this?

5. There seems to be a subnetwork displacement occurring along with the redox reaction, originated from the elimination of possible H-bonding interaction. This could be the probable reason behind the change in lattice parameter of the unit cells.

6. SEM imaging in fig. S4 shows rough edges for the 3D-TPB-COF-HQ(R) particles over the pristine 3D-TPB-COF-HQ particles, along with particle size lowering. Does it refer to the chemical instability of imine linked framework under subsequent redox cycles?

7. A tabular representation of N₂/CO₂ separation efficiency for previously reported materials is required to show the standpoint of current system against known materials.

Thus, this communication lacks of novelty to meet the high standard of Nature Communication and unfortunately, I do not find this suitable for publication. Moreover, the authors have not cited those above-mentioned reports which are closely related to this current work and are crucial to determine the novelty of the work by potential reviewer and reader.

Reviewer #3 (Remarks to the Author):

In this manuscript, by introducing hydroquinone unit into their key building block, Wang, Sun and coworkers demonstrated the first redox-triggered switchable three-dimensional (3D) COF. Such redox-triggered switching process was well-investigated by various techniques such as solid-state ¹³C NMR and PXRD. More impressively, accompanied with such switching process, the gas separation property of the 3D COFs could be well tuned attributed to the modulation of both the functional groups on the pore surface and the shape of pore channel. The authors described in this manuscript a very nice example of the construction of novel smart materials with tunable properties. I strongly recommend it to be published in Nature Communications after minor revision:

1. The yields of some key processes such as the reduction of 3D-TPB-COF-Q, the redox process of the model compound TPB-HQ, and the COFs hydrolysis are very important parameters that evaluate the efficiency of such redox switching strategy. However, currently, these data are missing;

2. The condensation product of TPB-HQ with aniline, rather than TPB-HQ itself, might be more suitable to serve as the model compound to investigate the redox-responsive property of the corresponding 3D COFs because the stability of the formed imine links in the 3D COFs should be also considered. The proper explanation should be provided ;

3. For the cycling experiments of the redox-switching of 3D COFs, the authors claimed such process could be repeated for at least 3 times. However, as shown in Supplementary Figure 15, in the ¹H NMR spectra of the second and third cycle (e and f), some obvious residue peaks between 7.5 and 8.0 ppm were observed. Does this mean that the reduction efficiency was decreased in the second and third cycles? The proper explanation should be provided. In addition, is there any other techniques, such as UV-vis spectroscopy, that can be used to directly evaluate such redox efficiency?

4. In this manuscript, the switching process was triggered by chemical redox. Have the authors tried to use electrochemical redox to drive the switching process? Compared with the chemical redox, the latter one might be a greener process.

Point-By-Point Response to the Reviewers' Comments

Reviewer #1:

Comment 1: Overall, this is a very interesting and novel manuscript that would be of interest to the readers of Nature Communications. Therefore, I recommend it for publication after the authors address the following.

Response: We thank the reviewer for the very positive comments.

Comment 2: Figure S1 depicts the FT-IR of both 3D-TPB-COF-HQ and 3D-TPB-COF-Q. Since this figure is used to demonstrate the formation of the imine-linked framework, I believe it would be beneficial to include the FT-IR spectra of the two monomers to further drive home the successful imine condensation. This figure is also used to observe the oxidation of 3D-TPB-COF-HQ to 3D-TPB-COF-Q, as noted by the appearance of a carbonyl stretch in the oxidized product. The spectra are cut off and have an upward bound of ~ 2500 wavenumbers. I believe it would be best if the O-H region was included to show the initial presence of O-H groups in the reduced form and their disappearance upon oxidation (this should also be applied to Figure S12).

Response: We thank the reviewer for bringing this point to our attention. We have performed the FT-IR experiment of two monomers (TPB-HQ and TPB-Q) and TAPM. Obviously, the formation of imine bonds was clearly confirmed. We have modified the corresponding Figure in the revised SI as Supplementary Figure 1.

Figure 1 FT-IR spectra of TAPM (black curve), TPB-HQ (red curve), TPB-Q (blue curve), 3D-TPB-COF-HQ (green curve) and 3D-TPB-COF-Q (purple curve).

We did record the FT-IR spectrum up to 4000 cm⁻¹. Unfortunately, even we performed the FT-IR experiments for several times, it is very difficult for us to get rid of the water and a broad peak centered at 3300 cm⁻¹ was found, due to the presence of

moisture. Therefore, we cannot distinguish the peak of the O-H groups in hydroquinone units in FT-IR spectra. To our delight, we have observed the characteristic peak of C=O groups in quinone units at about 1650 cm^{-1} , which can confirm the redox process. In addition, we recorded the ^1H NMR of digested COFs, which also confirmed the formation of reduced form. Moreover, this redox process was clearly confirmed by ssNMR and PXRD experiment. Therefore, for Supplementary Figure 12 (now Figure 13), in order to clearly show the changes of the characteristic peak at about 1650 cm^{-1} , we still set the upper bound of the spectra to 2500 cm^{-1} .

Comment 3: *Did the authors try any other oxidizing/reducing agents? Why did they use a different oxidizer for the polymer [BQ] vs. the small molecule [Fe(ClO₄)₃]?*

Response: We thank the reviewer for this comment. Actually, the TPB-HQ can be oxidized into TPB-Q after being treated with many oxidants, such as $\text{Na}_2\text{S}_2\text{O}_8$, $\text{Fe}(\text{ClO}_4)_3$, tetra-chloro-benzoquinone and *p*-benzoquinone [BQ]. As shown in supplementary scheme 2 and 4, we chose both $\text{Fe}(\text{ClO}_4)_3$ and BQ as the oxidant, with quantitative yield. However, for oxidation of 3D-TPB-COF-HQ, we choose BQ (not $\text{Fe}(\text{ClO}_4)_3$) as the oxidant, as we are worried about that the pore structure may absorb the remaining $\text{Fe}^{3+}/\text{Fe}^{2+}$ ion, which can further affect the adsorption properties of 3D-TPB-COF-Q.

Comment 4: *Since the authors mention that one advantage to using COFs over MOFs is the increased chemical stability of COFs, I believe that stability studies of the two synthesized 3D-COFs should be included in the SI.*

Response: We thank the reviewer for bringing this point to our attention. We have investigated the stability of these two 3D COFs in different environments. After immersing them into different solvents for 24 h, the PXRD spectra were recorded. Obviously, these two 3D COFs can maintain the crystallinity in these conditions. We have added this result into the revised SI as Supplementary Figure 6.

Figure 2 PXRD patterns of (a) 3D-TPB-COF-HQ and (b) 3D-TPB-COF-Q after treatment in different solvents for 24 hours.

Comment 5: In the manuscript, the authors claim that they are able to repeat the switching process three times “without losing crystallinity and porosity.” While they include data to support the retention of crystallinity, they do not include any data regarding the porosity of the material after repeated redox-switching.

Response: We thank the reviewer for bringing this point to our attention. We have performed the nitrogen sorption isotherm on 3D-TPB-COF-HQ(R) after three redox cycles at 77 K. The BET surface area of 3D-TPB-COF-HQ(R) was calculated to be 760 m² g⁻¹, indicating the retention of porosity. We have added this result into the revised SI as Supplementary Figure 17.

Figure 3 N₂ adsorption–desorption isotherm of 3D-TPB-COF-HQ(R) after three cycles at 77 K (a) and its pore size distribution (b).

Reviewer #2:

Comment 1: The manuscript submitted to *Nat. Commun.* by Gao et al. reports the synthesis a 3-dimensional covalent organic framework (COF) and its redox switching property by addition of oxidizing/reducing agent. The authors have tried to show the existence of the two forms through XRD analysis and advanced technique like electron diffraction.

Response: We thank the reviewer for this comment.

Comment 2: In the introduction section, they highlighted that COFs are superior materials than MOFs in terms of chemical stability, which is partly true. There are certain MOFs which are quite stable and the same group has already reported this redox switching phenomenon using a chemically stable Zr-based MOF (*Chem. Mater.* 2015, 27, 18, 6426-6431). The chemical stability of imine-based COFs is also not good, owing to the reversible nature of the condensation reaction.

Response: We thank the reviewer for this comment but we don't fully agree with the reviewer on this point. First, it should be pointed out here, we didn't claim COFs are superior material than MOFs in terms of chemical stability. In our manuscript, we mentioned “COF can show enhanced stability in contrast to **most** MOFs, due to their

robust covalent linkages in the framework". I believe this reviewer will agree, several types of MOFs (e.g. ZIF, MIL, etc.) are stable, but most of them are not.

Second, we did report a Zr-based MOF (UiO-68-OH) and investigate its switching behavior. Actually, in the experiment section of that paper, we mentioned UiO-68-OH crystals are always kept in the solvent for redox switching study. UiO-68-OH is stable by keeping in common solvents without the activation process (dried from solvents), but the activated UiO-68-OH can't retain the crystallinity after being re-immersed into most of the common solvents except CH₃CN (see Figure 4). For imine-linked 3D-TPB-COF-HQ, however, it is stable under the same conditions (see Figure 2a). Therefore, our new system showed certain superiorities in stability to the reported UiO-68-OH.

Figure 4 The PXRD patterns of activated UiO-68-OH before and after immersing in different solvents for 24 hours.

Comment 3: *Added to that, switching between hydroquinone/quinone moieties inside a 2-dimensional COF is already reported (Chem. Mater. 2017, 29, 5, 2074-2080), using more facile approach of electrochemistry. Although this communication attempted to emphasize on introduction of 3-dimensional framework for better applicability, 5-fold interpenetration of the nets leaves with minimal porosity at the end.*

Response: We thank the reviewer for this comment but we don't fully agree with the reviewer on this point. Indeed, the switching between hydroquinone/quinone moieties inside a 2D COF is already reported. However, we should emphasize that 2D COFs and 3D COFs are different, and we have also demonstrated that 3D COFs can show some unique advantages compared to 2D COFs (*ACIE*, 2020, 59, 3624). In fact, in COFs area, most works are related to 2D structures, but 3D COFs are much less investigated (around ~60 papers since 2007). Fundamentally, such underdeveloped status can be attributed to the existing challenges in 3D COFs, including synthetic difficulty, complicated structural determination, very few network topologies and limited building blocks. In addition, as a novel type of organic porous material, the functionalization of 3D COFs for interesting applications remains largely unexplored.

Therefore, the development of a stimuli-responsive 3D COF is very important for expanding their structural diversity and new application. In this work, we reported the first stimuli-responsive 3D COF, which can undergo a reversible transformation under stimuli, even the pore size is not very big. Moreover, their switching process can reversibly change the pore environments and thus endow the designed 3D COFs with interesting stimuli-responsive properties. According to this work, we strongly believe 3D COFs can provide robust platforms for efficient tuning of molecular switches in solid state. More importantly, switching of these moieties in 3D COFs can enable the resulting materials with interesting stimuli-responsive properties. From this result, we believe more stimuli-responsive 3D COFs that can respond to different stimuli (e.g. light, temperature, etc.) for interesting applications can be expected.

Comment 4: *Do the materials show any distinct emission property? If so, then monitoring the emission properties could be used to study the conversion kinetics between quinone/hydroquinone forms of the framework.*

Response: We thank the reviewer for this comment. Unfortunately, both COFs are nonfluorescent, and thus we cannot study the conversion kinetics by monitoring the emission properties.

Comment 5: *There is slight difference between adsorption and desorption curve for N₂ adsorption isotherm at 77 K for the hydroquinone moiety containing framework, but not observed with the quinone form. What could be the origin for this?*

Response: We thank the reviewer for this comment. We think the slight difference between adsorption and desorption curve for 3D-TPB-COF-HQ could be due to the small structure changes during gas adsorption and such structure changes could be the rotation of the OH groups or structure relaxation in 3D-TPB-COF-HQ. Such minor changes are difficult to be confirmed by simple experiments and is not the focus of our present studies.

Comment 6: *There seems to be a subnetwork displacement occurring along with the redox reaction, originated from the elimination of possible H-bonding interaction. This could be the probable reason behind the change in lattice parameter of the unit cells.*

Response: We thank the reviewer for bringing this point to our attention. We have measured all the possible hydrogen bond interaction position in 3D-TPB-COF-HQ. Among them, the shortest distance is 3.67 Å (-OH---N), which is too long for hydrogen bond interaction (Steiner, T. *Angew. Chem. Int. Ed.* **2002**, *41*, 48). Therefore, we don't think there is H-bonding interaction in 3D-TPB-COF-HQ.

Comment 7: *SEM imaging in fig. S4 shows rough edges for the 3D-TPB-COF-HQ(R) particles over the pristine 3D-TPB-COF-HQ particles, along with particle size lowering. Does it refer to the chemical instability of imine linked framework under subsequent redox cycles?*

Response: We thank the reviewer for this comment. According to the PXRD and absorption experiments, the crystallinity and porosity of 3D-TPB-COF-HQ was well-maintained during the redox process. Consequently, the framework is very stable under redox cycles. The change in the SEM image of the 3D-TPB-COF-HQ(R) may be ascribed to the crystal rupture by stirring force during the reaction. In addition, from the SEM image (Supplementary Figure 4a), the morphology of 3D-TPB-COF-HQ is not uniform and also have shown some particles.

Comment 8: A tabular representation of N₂/CO₂ separation efficiency for previously reported materials is required to show the standpoint of current system against known materials.

Response: We thank the reviewer for this comment. The adsorption selectivity for CO₂/N₂ compared to the other COFs was listed below. Obviously, our system is comparable or better than other systems. More importantly, we should emphasize here, this is the first time to show stimuli-responsive separation effect in 3D COFs.

Table 1 The adsorption selectivity for CO ₂ /N ₂ compared to the other COF systems					
3D COFs	Temp.	CO ₂ /N ₂ (v/v)	Method	Selectivity	Reference
3D-TPB-COF-HQ	273K	15:85	IAST	40	This work
3D-TPB-COF-Q	273K	15:85	IAST	93	
3D-IL-COF-1	298K	50:50	Henry	24.6	J. Am. Chem. Soc. 2018 , 140, 4494.
3D-IL-COF-2	298K	50:50	Henry	24.0	
3D-IL-COF-3	298K	50:50	Henry	24.4	
3D-COF-1a	298K	50:50	Henry	7.1	
3D-IL-COF-1b	298K	50:50	Henry	43.6	
CD-COF-Li	273K	15:85	IAST	106	Angew. Chem. Int. Ed. 2017 , 56, 16313
CD-COFDMA	273K	15:85	IAST	83	
CD-COF-PPZ	273K	15:85	IAST	86	
3D-Py-COF	273K	15:85	IAST	22.2	J. Am. Chem. Soc. 2016 , 138, 3302.
3D-TPB-COF-H	273K	15:85	IAST	24	Angew. Chem. Int. Ed. 2019 , 58, 9770
3D-TPB-COF-Me	273K	15:85	IAST	31	
3D-TPB-COF-F	273K	15:85	IAST	50	
[HO]100%-H ₂ P-COF	298K	15:85	IAST	8	Angew. Chem. Int. Ed. 2015 , 54, 2986
[HO ₂ C]100%-H ₂ P-COF	298K	15:85	IAST	77	
CAA-COF-1	273K	10:90	IAST	83	CrystEngComm , 2018 , 20, 7621
CAA-COF-2	273K	10:90	IAST	64	
TaPa-1	273K	10:90	IAST	53	
TpBd	273K	10:90	IAST	44	
Por-sp ² c-COF	273K	15:85	IAST	51	Angew. Chem. Int. Ed. 2019 , 58, 6430
JUC-505	273K	15:85	Henry	97.2	Nat. Chem. 2019 , 11, 587
JUC-506	273K	15:85	Henry	69	

Comment 9: *Thus, this communication lacks of novelty to meet the high standard of Nature Communication and unfortunately, I do not find this suitable for publication. Moreover, the authors have not cited those above-mentioned reports which are closely related to this current work and are crucial to determine the novelty of the work by potential reviewer and reader.*

Response: We thank the reviewer for this comment but we don't agree with the reviewer on the point.

COFs represent an emerging class of crystalline polymers with high porosity and structural tunability. Unlike the relatively well-established 2D COFs, 3D COFs are much less reported (~60 papers in total since first reported in 2007). In our opinion, all the people (not just us) working on 3D COFs will definitely admit that it is very challenging to design and synthesize novel 3D COFs. It should be emphasized here, the development of 3D COFs is in the early stage and their applications are far-away unexplored.

Stimuli-responsive materials have gained intensive attention over the past decades and shown great potentials in many areas. Porous materials have provided a promising platform to construct stimuli-responsive material, such as MOFs. However, the moderate stability of MOFs will be a fatal obstacle to the development for their practical use as stimuli-responsive porous materials. Therefore, considering that COFs can show enhanced stability in contrast to most MOFs due to their robust covalent linkages in the framework, we designed and synthesized a redox-responsive 3D COF (3D-TPB-COF-HQ), which can undergo a reversible transformation through a hydroquinone/quinone redox reaction. We have several novel points here:

1. 3D-TPB-COF-HQ is the first stimuli-responsive 3D COFs. By using several techniques, we clearly show that the switching process gradually happened through the COF framework, with an almost quantitative conversion yield. This study strongly demonstrates 3D COFs can provide robust platforms for efficient tuning of molecular switches in solid state.

2. The structure determination of 3D COFs has proven to be a big challenge. We successfully determined the structures of 3D-TPB-COF-HQ and 3D-TPB-COF-Q at atomic level, with a resolution up to 1.0 angstrom. With these high quality data, the detailed information of their structure changes in this transformation process is obtained.

3. The redox-triggered switching in 3D COFs will change the pore environment, which can result in tunable gas separation property. Consequently, we demonstrated the first example in 3D COFs with a remarkable stimuli-responsive separation effect.

This study strongly demonstrates 3D COFs can provide a suitable platform for efficient tuning of stimuli-responsive switches in solid state. More importantly, switching of these moieties in 3D COFs can remarkably modify the internal pore environment, which will thus enable the resulting materials with interesting stimuli-responsive properties. We think this is just a start and from this work, more stimuli-responsive 3D COFs that can respond to different stimuli (e.g. light, temperature, etc.) for interesting applications can be expected. Therefore, we strongly believe this work is novel enough and can be published in *Nature Communication*, which was also supported by reviewer 1 and reviewer 3.

We have added the following references into the revised manuscript:

14. Gui, B. et al. Reversible Tuning Hydroquinone/Quinone Reaction in Metal–Organic Framework: Immobilized Molecular Switches in Solid State. *Chem. Mater.* **27**, 6426–6431 (2015).

40. Chandra, S. et al. Molecular Level Control of the Capacitance of Two-Dimensional Covalent Organic Frameworks: Role of Hydrogen Bonding in Energy Storage Materials. *Chem. Mater.* **29**, 2074–2080 (2017).

Reviewer #3:

Comment 1: In this manuscript, by introducing hydroquinone unit into their key building block, Wang, Sun and coworkers demonstrated the first redox-triggered switchable three-dimensional (3D) COF. Such redox-triggered switching process was well-investigated by various techniques such as solid-state ¹³C NMR and PXRD. More impressively, accompanied with such switching process, the gas separation property of the 3D COFs could be well tuned attributed to the modulation of both the functional groups on the pore surface and the shape of pore channel. The authors described in this manuscript a very nice example of the construction of novel smart materials with tunable properties. I strongly recommend it to be published in Nature Communications after minor revision.

Response: We thank the reviewer for these very positive comments.

Comment 2: The yields of some key processes such as the reduction of 3D-TPB-COF-Q, the redox process of the model compound TPB-HQ, and the COFs hydrolysis are very important parameters that evaluate the efficiency of such redox switching strategy. However, currently, these data are missing;

Response: We thank the reviewer for bringing this point to our attention. In the revised Supplementary Information, we have added the reduction yield of 3D-TPB-COF-Q (92%). There has some weight loss, as some powders on the filter paper cannot be obtained. Therefore, we collected all the residues of digested COF sample for ¹H NMR study, which confirmed the quantitative conversion.

For the oxidation of TPB-OH, as there are some oxidant left, we didn't purified the compound by column since we are afraid the weight loss in the purification process. Therefore, we collected all the residues for ¹H NMR study, which showed quantitative transformation. For the reduction of TPB-Q, we have added the yield (85%) in the SI. There also has some weight loss, since some powders are left on the filter paper.

Comment 3: The condensation product of TPB-HQ with aniline, rather than TPB-HQ itself, might be more suitable to serve as the model compound to investigate the redox-responsive property of the corresponding 3D COFs because the stability of the formed imine links in the 3D COFs should be also considered. The proper explanation should be provided;

Response: We thank the review for this comment. First, we agree that the condensation

product of TPB-HQ with aniline should be more suitable to serve as the model compound to investigate the redox-responsive property. However, based on our experiences (we have been working on 3D imine-linked COFs for several years, see: *JACS*, **2016**, *138*, 3302; *JACS*, **2017**, *139*, 8705; *Nat. Commun.* **2018**, *9*, 5234; *ACIE*, **2019**, *58*, 9770; *JACS*, **2020**, *142*, 3718; *ACIE*, **2020**, *59*, 3624) and results from other groups, we are pretty sure the chemical stability of 3D imine-linked COFs are not bad, although the imine-linkage is not stable. Therefore, we used the precursor TPB-HQ to investigate the redox-responsive property. As expected, these COFs are indeed very stable in the oxidation and reduction process, at least for three times cycling.

Comment 4: For the cycling experiments of the redox-switching of 3D COFs, the authors claimed such process could be repeated for at least 3 times. However, as shown in Supplementary Figure 15, in the ^1H NMR spectra of the second and third cycle (e and f), some obvious residue peaks between 7.5 and 8.0 ppm were observed. Does this mean that the reduction efficiency was decreased in the second and third cycles? The proper explanation should be provided.

Response: We thank the reviewer for bringing this point to our attention and pointing out our mistake. We thought that the observed residue peaks at 7.75 ppm in the ^1H NMR spectra of the second and third cycle may be caused by improper experimental operation, since in the first cycle they were totally disappeared and all the quinone units in 3D-TPB-COF-Q were quantitatively reduced back to hydroquinone. Therefore, we performed the experiment again more carefully. The results showed that the reduction efficiency could be maintained in the second and third cycles. We have added this new figure into the revised SI as Supplementary Figure 16.

Figure 5 ^1H NMR spectra of digested powders during the reversible transformation in three oxidation/reduction cycles. (a) 3D-TPB-COF-HQ; (b) 1st oxidation; (c) 1st reduction; (d) 2nd oxidation; (e) 2nd reduction; (f) 3rd oxidation; (g) 3rd reduction.

Comment 5: *In addition, is there any other techniques, such as UV-vis spectroscopy, that can be used to directly evaluate such redox efficiency?*

Response: We thank the reviewer for this comment. In fact, as it is quite difficult for us to figure out the right condition for digesting COFs sample, we have considered other techniques. For example, we have considered UV-vis or fluorescent spectroscopy. Unfortunately, there is a big overlap between the UV-vis spectra of 3D-TPB-COF-HQ and 3D-TPB-COF-Q (Figure 6), and both COFs are nonfluorescent. Therefore, we think it is very difficult for us to use other techniques to evaluate the redox efficiency, except the ^1H NMR.

Figure 6 Solid-state UV-Vis absorption spectra of 3D-TPB-COF-HQ (black curve) and (b) 3D-TPB-COF-Q (red curve).

Comment 6: *In this manuscript, the switching process was triggered by chemical redox. Have the authors tried to use electrochemical redox to drive the switching process? Compared with the chemical redox, the latter one might be a greener process.*

Response: We thank the reviewer for this comment. In fact, we have recorded the CV spectra of the COFs for several time, but we didn't see apparent peaks. This may be explained by the low conductivity of 3D COFs, which is totally different from 2D COFs that has π - π interaction in the adjacent layer. We think it is possible to perform the electrochemical redox reaction by growing the thin-film of 3D-TPB-COF-HQ and 3D-TPB-COF-Q, but to be honest, this is a very tough task (we have been working on this for almost two years, but the progress is slow).

More importantly, as we mentioned, the main point of our paper is that we want to demonstrate 3D COFs can provide robust platforms for efficient tuning of molecular switches in solid state and the resulting materials can show interesting properties. Therefore, we investigated the redox-switching process with several techniques, including NMR, PXRD, Gas absorption, and rotation electron diffraction. From these detailed experiments, we can clearly conclude the switching process gradually happened through the COF framework, with an almost quantitative conversion yield. In addition, the redox-triggered switching in 3D COFs can change the pore environment, leading a remarkable stimuli-responsive separation effect. Based on this study, we

strongly believe more stimuli-responsive 3D COFs that can respond to different stimuli (e.g. light, temperature, etc.) for interesting applications can be expected. Indeed, the using of electrochemistry for redox switching is much greener, but we cannot get these detailed information.

REVIEWERS' COMMENTS:

Reviewer #1 (Remarks to the Author):

My comments have been addressed and I'm happy with the revised version of the manuscript.

Reviewer #2 (Remarks to the Author):

[Note from the Editor: Reviewer #2 made remarks to the editor only and leaves the decision to accept the manuscript to the editor]

Reviewer #3 (Remarks to the Author):

This is a revised manuscript submitted by Wang and coworkers. In this manuscript, they presented a very impressive example of construction of novel three-dimensional (3D) COFs with tunable properties. I have checked the updated version very carefully. I can tell that the authors have made very extensively revision according to all reviewers' suggestions, which obviously improve the quality of the manuscript. Considering the impressive novelty and significance presented in this manuscript, I strongly recommend the current version to be accepted by Nature Communications.

Reviewer #1:

Comment: My comments have been addressed and I'm happy with the revised version of the manuscript.

Response: We thank the reviewer for this positive comment.

Reviewer #2:

Comment: [Note from the Editor: Reviewer #2 made remarks to the editor only and leaves the decision to accept the manuscript to the editor]

Response: We thank the reviewer for the comments.

Reviewer #3:

Comment: This is a revised manuscript submitted by Wang and coworkers. In this manuscript, they presented a very impressive example of construction of novel three-dimensional (3D) COFs with tunable properties. I have checked the updated version very carefully. I can tell that the authors have made very extensively revision according to all reviewers's suggestions, which obviously improve the quality of the manuscript. Considering the impressive novelty and significance presented in this manuscript, I strongly recommend the current version to be accepted by Nature Communications.

Response: We thank the reviewer for these very positive comments.